# Retinal Blood Vessel Analysis Using Optical Coherence Tomography (OCT) in Multiple Sclerosis

**DOI:** 10.3390/diagnostics13040596

**Published:** 2023-02-06

**Authors:** Nicholas Young, Robert Zivadinov, Michael G. Dwyer, Niels Bergsland, Bianca Weinstock-Guttman, Dejan Jakimovski

**Affiliations:** 1Buffalo Neuroimaging Analysis Center (BNAC), Department of Neurology, Jacobs School of Medicine and Biomedical Sciences, University at Buffalo, State University of New York, Buffalo, NY 14260, USA; 2Center for Biomedical Imaging and Clinical Translational Science Institute, University at Buffalo, State University of New York, Buffalo, NY 14203, USA; 3RCCS, Fondazione Don Carlo Gnocchi ONLUS, 20121 Milan, Italy; 4Jacobs Comprehensive MS Treatment and Research Center, Department of Neurology, Jacobs School of Medicine and Biomedical Sciences, University at Buffalo, Buffalo, NY 14202, USA

**Keywords:** multiple sclerosis, optical coherence tomography, retinal vessels, retinal nerve fiber layer, vessel diameter, cardiovascular

## Abstract

*Background:* Both greater retinal neurodegenerative pathology and greater cardiovascular burden are seen in people with multiple sclerosis (pwMS). Studies also describe multiple extracranial and intracranial vascular changes in pwMS. However, there have been few studies examining the neuroretinal vasculature in MS. Our aim is to determine differences in retinal vasculature between pwMS and healthy controls (HCs) and to determine the relationship between retinal nerve fiber layer (RNFL) thickness and retinal vasculature characteristics. *Methods:* A total of 167 pwMS and 48 HCs were scanned using optical coherence tomography (OCT). Earlier OCT scans were available for 101 pwMS and 35 HCs for an additional longitudinal analysis. Segmentation of retinal vasculature was performed in a blinded manner in MATLAB’s optical coherence tomography segmentation and evaluation GUI (OCTSEG) software. *Results:* PwMS has fewer retinal blood vessels when compared to HCs (35.1 vs. 36.8, *p* = 0.017). Over the 5.4 year follow up, and when compared to HCs, pwMS has a significant decrease in number of retinal vessels (average loss of −3.7 *p* = 0.007). Moreover, the total vessel diameter in pwMS does not change when compared to the increase in vessel diameter in the HCs (0.06 vs. 0.3, *p* = 0.017). Only in pwMS is there an association between lower RNFL thickness and fewer retinal vessel number and smaller diameter (r = 0.191, *p* = 0.018 and r = 0.216, *p* = 0.007). *Conclusions:* Over 5 years, pwMS exhibit significant retinal vascular changes that are related to greater atrophy of the retinal layers.

## 1. Introduction

Multiple sclerosis (MS) is a chronic demyelinating and neurodegenerative disease of the central nervous system that is characterized by episodic neurological attacks of inflammation, axonal injury, and gliosis [1]. Multiple studies recently investigated the vascular contributions to the MS pathophysiology, and showed that the presence of cardiovascular comorbidities associated with poorer physical, cognitive, and clinical MS outcomes [2,3]. Moreover, the presence of cardiovascular comorbidities also contributes towards higher MS lesion burden and lead to a greater rate of brain atrophy [4]. A possible common etiological mechanism between MS and its vascular comorbidities has been proposed where the years around MS diagnosis are also accompanied with new onset of cardiovascular and cerebrovascular events [5]. The impact of cardiovascular comorbidities in people with MS (pwMS) is also linked with changes through the heart–brain axis. Several post-mortem and imaging studies demonstrate higher level of peripheral and central atherosclerosis in pwMS that lead to morphological vessel changes [6].

One useful technique for continuing the research into the cerebral vasculature and neurodegeneration of pwMS is through the use of optical coherence tomography (OCT) [7]. OCT is a fast five-min-long scan that can non-invasively provide an in-depth image of the retina that can allow both qualitative and quantitative investigation of the retinal layers [8]. It has been frequently utilized as a tool for monitoring retinal changes (macular edema during treatment with sphingosie-1-phosphate modulators) or as a proxy of the global neurodegenerative processes in pwMS. When compared to MRI, OCT provides a much more comfortable experience that can be utilized at each routine clinical visit. A systemic review and meta-analysis showed that the thinning of the retinal nerve fiber layer (RNFL) in pwMS was, on average, greater than the extent expected change in normal aging and was even more pronounced in pwMS presenting with optic neuritis (ON) [7].

The use of non-invasive OCT and OCT angiography (OCT-A) techniques over conventional vessel analysis methods are multifold. MRI-based studies such as MR angiography and conventional angiography requires the injection of gadolinium-based fluorescein and indocyanine green dyes that could cause anaphylactic reactions and other systemic adverse events [9]. Moreover, gadolinium-based contrast agents are limited to only subjects with healthy kidney function and repeated use of such agents can result in both systemic and cerebral contrast accumulation [10]. Recent segmentation methods allow retrospective analysis of already acquired A-image OCT scans, even when the OCT-A scan is not available [11].

The aim of this work is to determine if there are differences in retinal vasculature between pwMS and healthy controls (HCs) and to determine if there is a relationship between the retinal vessel characteristics with the volume of the retinal layers. We hypothesize that pwMS would exhibit altered vascular retinal structure that worsens over a mid-term period. We also hypothesize that these vascular retinal changes may be related to greater neurodegeneration within the retinal layers.

## 2. Materials and Methods

### 2.1. Study Population

The patients that participated in this work were a part of a larger, prospective study that examined cardiovascular, environmental, and genetic factors in multiple sclerosis (CEG-MS) variables in pwMS enrolled from 2014–2017 [8,12]. The inclusion criteria were as follows: (1) age between 18 and 75 years old; (2) having MRI, neuropsychological examination, and OCT scan within 30 days of the clinical examination; (3) being a HC required no known history of a neurological disorder; and (4) pwMS diagnosed as defined by the 2010—revised McDonald criteria [13]. The exclusion criteria were as follows: (1) known history of morphological vascular abnormalities (Klippel–Trenaunay–Weber, Parkes–Weber, Servelle–Martorell, or Budd–Chiari syndromes); (2) history of major depressive disorder, mood disorders, or other confirmed psychiatric diseases; and (3) pregnant and nursing mothers. The demographic and clinical data were collected using a previously published structured interview-based questionnaire and were cross-referenced with electronic medical records [4]. Among these variables, data regarding sex, age of symptom onset, annualized relapsing rate (ARR), and the status of disease-modifying therapy (DMT) were included. All interferon-β preparations were considered in one group, all generic and proprietary glatiramer acetate preparations were also considered as one group. Teriflunomide, dimethyl fumarate, and fingolimod were considered oral DMTs. Intravenous immunoglobulins, azathioprine, mitoxantrone, and mycophenolic acid were considered as off-label DMTs. MS patients were examined by an experienced neurologist and the Expanded Disability Status Scale (EDSS) was used to determine the level of physical disability [14]. Based on the clinical presentation and disease history, the pwMS were classified into relapsing–remitting MS (RRMS) and progressive MS (PMS) according to the 2013 Lublin criteria [15]. Due to the low individual sample size, the primary progressive MS (PPMS) and secondary progressive MS (SPMS) groups were merged. Only at the follow-up timepoint, the vision abilities were assessed using the low-contrast letter acuity (LCLA) test at 100% contrast, 2.5% contrast, and 1.25% contrast levels. The study was approved by the local Institutional Review Board (IRB) and all study participants signed written consent forms.

### 2.2. OCT Acquisition and Analyses

A total of 167 pwMS (113 pwRRMS and 54 pwPMS) and 48 HCs were scanned using OCT. Baseline OCT scans were retrieved from the baseline study timepoint and available in a smaller sample size of 101 pwMS and 35 HCs for an additional 5 year follow up analysis. OCT scans were completed on a Heidelberg Spectralis^®^ OCT machine (Heidelberg, Germany) without any pharmacological dilatation of the pupils in a dark room. Scans were determined to be of sufficient quality using the OSCAR-IB criteria. Scans with quality lower than 15 were excluded from the analysis. Each OCT acquisition is corrected for the differences derived from the subject’s refraction error. The retinal vasculature segmentation was performed on the peripapillary RNFL scans, a ring-type (3.3 mm and 12.0 circle diameter) B scan that was acquired using 768 A-scans. Real-time eye tracking mode was set on maximum 100 frames. These OCT scans were then imported into MATLAB’s optical coherence tomography segmentation and evaluation graphical user interface (OCTSEG.) The software is freely available at: https://www.mathworks.com/matlabcentral/fileexchange/66873-octseg-optical-coherence-tomography-segmentation-and-evaluation-gui (accessed 2 February 2023). Automated segmentations were run on all of the imported scans and then all scan segmentations were manually corrected and adjusted in a blinded manner. Both raters for the segmentation were not privy of the disease status and were not aware if the scan was from the baseline or follow-up visit. The first analysis was completed by the first rater and the second reproducibility analysis was completed by both raters again after one month. Once all OCT images were manually segmented without any particular order, and the results were extracted as comma separated values (CSVs). The process of OCT segmentation is shown in Figure 1 and Figure 2. Iterating pixel by pixel, the software produces a 0 to where there is no vessel present and a 1 to indicate the presence of a vessel highlight. Summation of all of highlighted pixels and multiplication by the pixel scale allowed calculation of the total vessel diameter. All vessels segmented using this methodology could be classified as part of the superficial capillary plexus. The segmentation process has been previously utilized in a similar publication shown elsewhere [16]. In particular, the peripapillary circle shown on the right-hand side of Figure 1 is unwrapped and each vertical line represents pixels of the circle. A short code within Spyder integrated development environment (Python platform) allowed quantification of the groupings of 1′s to calculate the total number of blood vessels in each OCT image. Lastly, we calculated the average diameter per vessel using the two aforementioned values. RNFL and ganglion cell inner plexiform layer (GCIPL) thicknesses were automatically segmented in the native Heidelberg software. The GCIPL thickness was determined on a macular level through a scan containing 61 B-scans sections (each containing 786 A scans) and real-time tracking set at 9 frames. A circular 1 mm/3 mm/6 mm early treatment diabetic retinopathy study (ETDRS) grid covered area of 30° × 25° with 120 μm spacing. Unfortunately, due to the old pattern of OCT data acquisition, we did not have peripapillary B-scan images that could allow quantification of both RNFL and GCIPL from the same retinal site.

### 2.3. Statistical Analyses

The statistical analysis was performed using SPSS 26.0 (IBM, Armonk, NY, USA.) Differences between demographic and retinal variables between pwMS, HC, pwRRMS, and pwPMS were calculated using chi-square test, Student’s t-test, Mann–Whitney U test, and analysis of covariance (ANCOVA) adjusted for age, as appropriate. In particular, the chi-square test was used to compare categorical variables such as sex ratios, history of ON, and disease phenotype. Student’s t-test was used to compare normally distributed numerical values such as age, disease duration, age of symptom onset, ARR, change in EDSS, and the retinal vessel and nerve layer characteristics. Due to the ordinal nature of the EDSS, a non-parametric Mann–Whitney U test was used. The associations between RNFL and vessel characteristics were analyzed using non-parametric Spearman’s correlations. The relationship between RNFL change over the follow-up with baseline and change in retinal vessel measures was assessed using linear regression model. The intra-rater and inter-rater reproducibility were assessed on a batch of 30 OCT scans and analyzed using intra-class correlation coefficient (ICC). *p*-values of less than 0.05 were considered statistically significant.

## 3. Results

### 3.1. Demographic and Clinical Characteristics of the Study Population

The demographic, clinical, and treatment characteristics of all pwMS, pwRRMS, pwPMS (7 PPMS and 38 SPMS), and HCs are shown in Table 1. The pwMS were, on average, 47.4 years old (SD = 11.5) with an average disease duration of 13.8 (SD = 9.9) years. There was not a significant difference in the sex ratio between pwMS and HCs. PwRRMS had a disease duration of 11.08 years and pwPMS had a disease duration of 21.00 years. PwPMS had a significantly greater EDSS both at baseline and follow up when compared to pwRRMS (median 5.7 vs. 1.5, *p* < 0.001 and median 6.5 vs. 2.0, *p* < 0.001, respectively). pwPMS had a larger annual relapse rate when compared to pwRRMS. There were no significant differences in the history of optic neuritis (*p* = 0.688). A significant difference in the pattern of DMT use among the pwRRMS and pwPMS was noted (*p* = 0.038). The pwMS had significantly worse performance when compared to the HCs in 100% contrast LCLA assessment (52.6 vs. 55.9, *p* = 0.001), 2.5% contrast LCLA assessment (36.9 vs. 43.2, *p* < 0.001), and 1.25% contrast assessment (29.8 vs. 36.0, *p* = 0.018). The pwPMS also had significantly worse vision performance when compared to the pwRRMS for both 100% contrast LCLA assessment (49.3 vs. 53.9, *p* = 0.002) and 2.5% contrast LCLA assessment (31.9 vs. 38.9, *p* = 0.004).

### 3.2. OCT-Based Measures at Baseline and Follow-Up

The intra-rater and inter-rater reproducibility of the retinal vasculature segmentation is shown in Table 2. The intra-rater reproducibility of the total vessel diameter and number of vessels are excellent (ICC 0.906, 95% CI 0.8–0.956) and good (ICC 0.762, 95% CI 0.493–0.888), respectively. Both measures demonstrate good inter-rater reproducibility as well (ICC 0.774, 95% CI 0.519–0.894 and ICC 0.876 95% CI 0.736–0.942 for total vessel diameter and number of vessels, respectively).

The characteristics of the retinal vasculature and retinal layers are shown in Table 3. At baseline and follow up, pwMS show a significantly smaller RNFL thickness compared to HCs (86.3 µm vs. 99.9 µm, *p* = 0.004 and 82.8 µm vs. 95.9 µm, *p* ≤ 0.001, respectively) and smaller GCIPL thickness (75.3 µm vs. 86.8 µm, *p* = 0.016 and 74.1 µm vs. 79.9 µm, *p* ≤ 0.001, respectively) while pwRRMS show a significantly larger RNFL thickness compared to pwPMS (88.6 µm vs. 79.4 µm and 86.1 µm vs. 74.1 µm, respectively). At baseline and follow up, HCs also show a significantly greater macular volume when compared to pwMS (8.8 mm^3^ vs. 7.9 mm^3^, *p* = 0.004 and 8.5 mm^3^ vs. 8.2 mm^3^, *p* = 0.001, respectively). Lastly, at follow up, the HCs display a significantly larger number of retinal vessels when compared to pwMS (36.8 vs. 35.1, respectively). Within the pwMS and HCs groups, there are no inter-eye differences in the number of retinal vessels (left eye 17.5 vs. right eye 17.5, *p* = 0.926, and left eye 18.6 vs. right eye 18.1, *p* = 0.354, respectively) and total vessel diameter (left eye 1.3 vs. right eye 1.2, *p* = 0.145, and left eye 1.3 vs. right eye 1.4, *p* = 0.597). Differences in number of retinal vessels between example cases of pwMS and HC are shown in Figure 3.

There are no differences in retinal vessel metrics at the follow-up between MSON and non-MSON eyes. When compared to the 214 non-MSON eyes, the 120 MSON-affected eyes of pwMS have similar per-eye total vessel diameter (1.2 vs. 1.3, *p* = 0.117), similar average number of vessels (17.2 vs. 17.7, *p* = 0.182), and similar average per-vessel average diameter (0.07 vs. 0.07, *p* = 0.877).

Table 4 shows the changes in retinal vasculature characteristics in pwMS, HCs, pwRRMS, and pwPMS over the 5 year follow up. PwMS show a significantly smaller increase in vessel diameter and a significant reduction in the number of retinal blood vessels when compared to HCs (0.06 cm vs. 0.3 cm, *p* = 0.017 and −3.7 vs. 0.7, *p* = 0.007, respectively). Despite the changes in number and total vessel diameter, the average per-vessel diameter between pwMS and HCs remains similar (increase of 0.008 vs. 0.007, *p* = 0.661). In particular, the decrease in number of vessels over the follow-up has weak correlation with decrease in total vessel diameter (r = 0.295). No differences in the change in retinal vessel architecture between the pwRRMS and pwPMS is notable in the change in total vessel diameter, total number of vessels, or in the average per-vessel diameter.

### 3.3. Relationship between Retinal Vasculature and Retinal Layer Thickness in the Study Population

The relationship between retinal vasculature and retinal thickness is shown in Table 5. It is only in pwMS that there is a correlation between total vessel diameter and the number of vessels with the RNFL thickness (r = 0.216, *p* = 0.007 and r = 0.191, *p* = 0.018, respectively). It is only in pwRRMS that there is a correlation with retinal vessel number and RNFL thickness (r = 0.221 and *p* = 0.025). Total vessel diameter shows a significant correlation with macular volume in pwMS, pwRRMS, and pwPMS (r = 0.259 and *p* = 0.001, r = 0.238 and *p* = 0.017, r = 0.335 and *p* = 0.017, respectively). It is only pwMS who show a significance in correlation between number of vessels and macular volume (r = 0.174 and *p* = 0.033). pwPMS are the only group to show a correlation with GCIPL thickness (r = 0.387 and *p* = 0.007) while pwMS and pwPMS show a significant association in number of retinal vessel and GCIPL thickness (r = 0.252 and *p* = 0.002 and r = 0.355 and *p* = 0.013, respectively). Scatter plots that provide visualization of the data dispersion and the correlations between the aforementioned measures are shown in Figure 4.

In the pwMS group, the percent change in RNFL over the follow-up is significantly associated with all three changes in retinal vasculature. After adjusting for age, greater rate of RNFL atrophy is associated with greater absolute increase in vessel diameter (standardized beta = −1.007, *p* = 0.009), decrease in number of vessels (standardized beta = 1.109, *p* = 0.045), and decrease in average per-vessel diameter (standardized beta = 0.814, *p* = 0.025). Baseline retinal vasculature measures are not associated with the change in RNFL. The discrepancies regarding the concurrent retinal vessel change and percent RNFL change are discussed in the limitations section of the manuscript.

Lastly, a higher number of vessels in the pwMS group is significantly associated with better vision as assessed through the 100% contrast LCLA test (r = 0.188, *p* = 0.033). This association is driven by the larger pwRRMS group, where a higher number of retinal vessels is associated with better 100% contrast LCLA performance (r = 0.262, *p* = 0.012).

## 4. Discussion

In this longitudinal OCT-based analysis of the retinal architecture, pwMS demonstrate significantly lower number of retinal vessels when compared to age-matched HCs. The semi-automated retinal vessel analysis was reproducible and fairly easy to perform. Over the 5 year follow-up, pwMS demonstrate significant loss in number of retinal vessels, whereas the HCs do not. The lower number of vessels and smaller retinal vessel diameter are associated with greater retinal layer atrophy as measured by the thickness of RNFL and GCIPL. Lastly, the changes in retinal vasculature may be related to the visual performance measured in pwMS. Of particular note, the associations in our study are of weak to moderate size and should be additionally replicated in other MS cohorts. The implications of these findings and the comparison to the literature are discussed hereafter.

The retinal changes evidenced in our study have been reported even at the very start of MS disease. An OCT angiography analysis shows that people with an initial demyelinating attack have significantly smaller vessel density in the radial peripapillary capillary plexus when compared to HCs [17,18]. After 2 years of follow-up, pwMS also show a significant decrease in vessel density in the superficial and deep capillary plexi [18]. We further corroborate the loss of vessel numbers in a much older population with long-standing MS. The extent of retinal vascular pathology was also shown in a recent Norwegian study where 23 newly diagnosed and untreated pwMS had significantly smaller retinal venular and arteriolar total diameter when compared to matched 23 HCs. [19] While these anatomical changes did not influence the oxygen saturation, these pwMS reported significantly worse low-contrast acuity performance [19]. A recent and much larger OCT-A study of 111 pwMS suggested that lower density only in the superficial retinal vascular plexus was associated with MS-based disability scores [20].

Outside of the prototypical retinal layer atrophy due to optic neuritis, the retinal involvement in pwMS can include a plethora of ocular manifestations [21]. Uveitis can be the first presenting symptom in up to a third of pwMS, and MS is a recognized non-infectious etiology by the working group for Standardization of Uveitis Nomenclature (SUN) [22,23]. Importantly, expression of the HLA-DRB1*15:01 haplotype is commonly associated with both diseases [24]. Long-standing and underappreciated ocular processes can lead to occlusive vasculitis or continuous narrowing of the retinal vessels. In a similar fashion, previous studies reported significant prevalence of primary retinal inflammation and retinal periphlebitis, an inflammatory vasculitis that affects the peripheral retina, in pwMS [25]. Moreover, the presence of retinal phlebitis has been correlated with future disease progression and higher lead of MRI disease burden (higher lesion volume and lower whole brain volume) [25]. Adaptive optics scanning laser ophthalmoscopy can also be used to visualize the abundant vascular and paravascular retinal changes that are present in either ON and non-ON MS eyes [26]. The MS disease itself, inflammatory eye changes, and cardiovascular comorbidities also share the same inflammatory cytokines (IL-1, IL-6, IL-17, TNF-α) that can propagate further worsening of the retinal health.

The vascular changes in the retina of pwMS could potentially be compared to other extracranial vascular changes previously evidenced [27,28]. The decrease in number of neck vessels over a similar timeframe was seen in pwMS and not in HCs. Depending on which cervical level the vessels are measured, pwMS tend to lose between 3 and 5 vessels over 5 years [29]. Moreover, the cross-sectional analysis demonstrates that pwMS initially have a greater number of collateral vascularization with a significantly greater number of vessels in the neck when compared to HCs [27]. These seemingly discrepant findings can be explained by the effect of inflammatory cytokines that dynamically change over the course of MS. The initial disease is highlighted by greater inflammatory phase where MS-specific activated CD4 lymphocytes also secrete vascular endothelial growth factor (VEGF) and induce neovascularization [30,31]. Cerebral inflammation can also result with significantly increased blood flow. As the inflammatory phase of MS winds down, the inflammatory drivers of neovascularization significantly decrease and may result in a steep decline in the number of vessels. MS inflammation was also recently associated with a greater rate of vessel remodeling, which includes fibrillar collagen type I deposition, luminal enlargement, and thickening of the perivascular space [32]. Our findings also corroborate previous studies indicating that pwMS experience much greater atherosclerotic burden that results in the narrowing of the arterial lumen and greater plaque formation [33]. The significant increase in total retinal vessel diameter over the follow-up in the pwMS could potentially be explained by the loss of smaller vessels to arteriosclerotic processes, which prompts greater vasodilatation in the remaining vessels. Interestingly, the increase in retinal vessel diameter is suggested as one of the earliest signs of diabetic retinopathy. The gene-induced intra-individual and intra-familial co-occurrence of type 1 diabetes and MS is already known [34]. Moreover, the accrual of physical disability in pwMS could lead to poorer lifestyle, lack of exercise, and development of metabolic syndrome. Lastly, the reduction in small vessel density in pwMS is also seen in the lesional and periventricular white matter [35]. The correlation between white matter venous density and whole brain volume also indicates the relationship between neurodegenerative features and vascular changes [35].

Our study does have several limitations that should be considered. Although reproducible in our hands, the current protocol of a semi-automated segmentation of the retinal vessels can be operator-dependent and potentially biased. In our experience, the automated and non-corrected segmentation largely fails at producing meaningful data in scans of relatively lower acquisition quality. Future improvement of the segmentation process through implementation of artificial intelligence (AI) methods can substantially diminish the draw-backs of a semi-automated analysis. In order to circumvent these limitations, our study implemented strict blinding and randomization, where both operators were not aware of (1) disease status and, more importantly, (2) the study timepoint (baseline or follow-up scan). Moreover, the paired scans were also randomized and not analyzed consecutively (not biasing the vessel segmentation due to recent recollection of the retinal architecture from the other timepoint). That said, the limitations derived from human interaction remain an essential bias in extending such analysis in any routine use. Concurrent use of B-scans and OCT angiography could provide the much-needed criterion validity. The discrepancy between the number of vessels, the total vessel diameter, and average per-vessel diameter further limit the generalizability of the findings. For example, some pwMS have a loss of more than 10 vessels and stable total diameter (increase in compensatory per vessel diameter), while other pwMS have an increase in the number of vessels and decrease in total diameter (more but smaller vessels). When these pwMS are averaged as part of a group comparison, the differences would be nullified. Development of change patters that could incorporate different directionality changes could improve the overall associations between retinal vasculature measures with clinical and phenotypical outcomes. Another limitation in our analysis is the lack of retinal perfusion measures and retinal oximetry. As it stands, the retinal layer atrophy may be a result of either vessel-independent trans-synaptic nerve changes in the optic nerve or due to local hypoperfusion. Lastly, the RNFL and GCIPL quantification is acquired on two separate retinal areas. Future quantification of GCIPL in B-scan images of the peripapillary area could circumvent this limitation.

In conclusion, quantification of the retinal vascular characteristics on a peripapillary OCT scan is repeatable and reproducible. PwMS have significantly fewer retinal vessels when compared to age-matched HCs. Over 5 years follow-up, pwMS continue to lose significantly more retinal vessels when compared to controls. Lastly, fewer retinal vessels and smaller total vessel diameter are associated with thinner RNFL and GCIPL. Over follow-up, the change in retinal vasculature in pwMS has been associated with greater retinal atrophy.

## Figures and Tables

**Figure 1 diagnostics-13-00596-f001:**
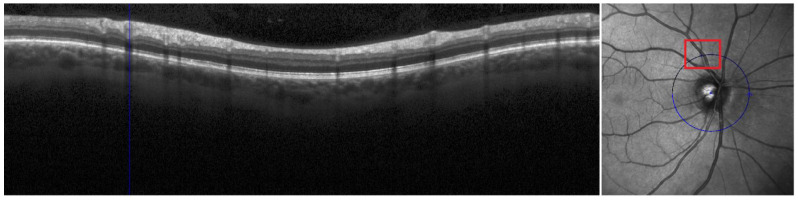
Layout of the OCT image in the OCTSEG software. The blue circle on the right-hand image is unwrapped and showcased on the left side. The blue line shown in the left side of the picture is indicating the vessel that is highlighted with red square.

**Figure 2 diagnostics-13-00596-f002:**
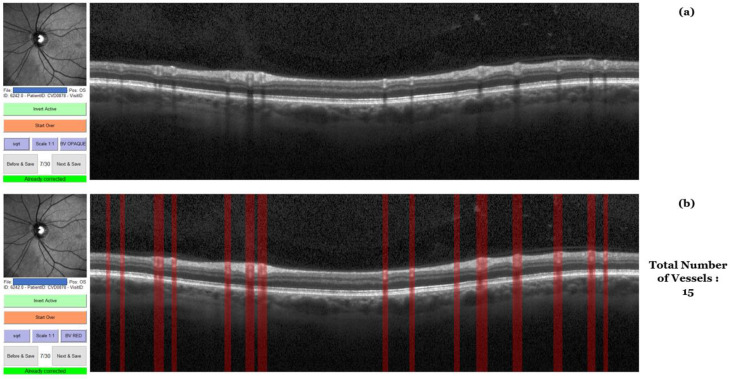
An example of manual OCT segmentation in the OCTSEG software. (**a**) An OCT image imported into the OCTSEG software with no segmentation. (**b**) A figure of the OCT shown in (**a**) but after manual segmentation. The segmentations are represented as red columns spanning the diameter of the blood vessel. Our code returns a total of 15 blood vessels in this example.

**Figure 3 diagnostics-13-00596-f003:**
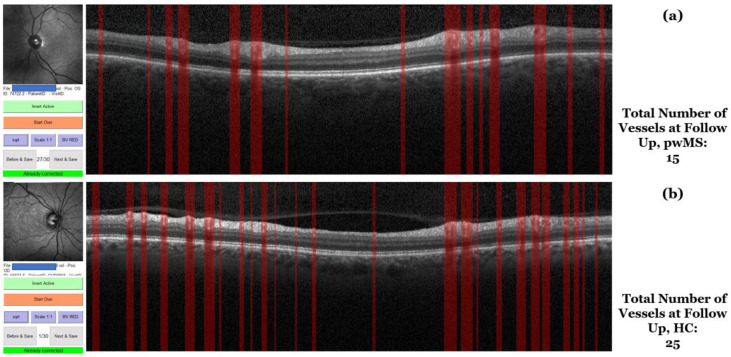
Comparison of an OCT of a pwMS and a HC at 5 year follow up. **Legend**: pwMS—person with multiple sclerosis, HC—healthy control, OCT—optical coherence tomography. (**a**) A figure showing the segmentation of the pwMS’s OCT. (**b**) A figure showing the segmentation of the HC’s OCT at 5 year follow up.

**Figure 4 diagnostics-13-00596-f004:**
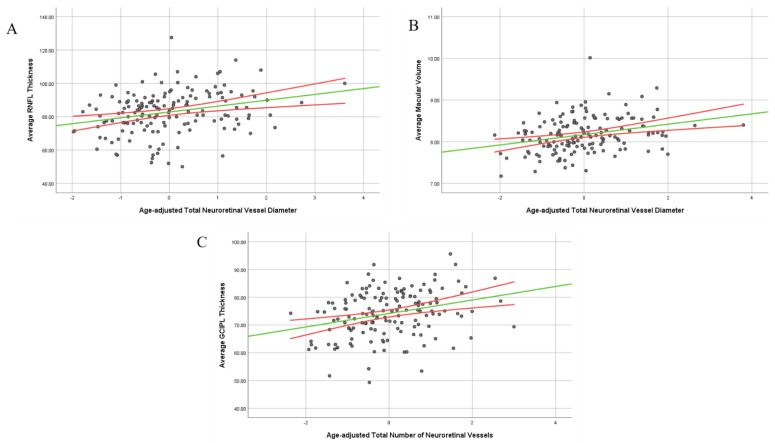
Age-adjusted correlations in pwMS between neuroretinal vascular measures and anatomical retinal layers. pwMS—people with multiple sclerosis, RNFL—retinal nerve fiber layer, MV—macular volume, GCIPL—ganglion inner plexiform layer. (**A**) Age-adjusted relationship between average RNFL thickness and total neuroretinal vessel diameter, (**B**) age-adjusted relationship between average MV and total neuroretinal vessel diameter, (**C**) age-adjusted relationship between average GCIPL and total number of neuroretinal vessels.

**Table 1 diagnostics-13-00596-t001:** Demographic and clinical characteristics of the study population at baseline visit.

Demographics Characteristics	pwMS (*n* = 167)	pwRRMS (*n* = 122)	pwPMS (*n* = 45)	HCs (*n* = 48)	pwMS vs. HCs *p*-Value	pwRRMS vs. pwPMS *p*-Value
Female, n (%)	122 (73.1)	86 (70.5)	36 (80.0)	33 (68.8)	0.558	0.402
Age in years, mean (SD)	47.4 (11.5)	44.4 (11.2)	55.5 (7.9)	43.75 (14.8)	0.125	**<0.001**
Time of follow-up, mean (SD)	5.5 (0.6)	5.5 (0.6)	5.4 (0.6)	5.6 (0.5)	0.293	0.76
Disease duration, mean years (SD)	13.8 (9.9)	11.08 (8.4)	21.00 (10.2)	-	-	**<0.001**
Age of symptom onset, mean (SD)	33.45 (10.1)	33.08 (9.7)	34.44 (11.3)	-	-	0.442
RRMS/PMS, n (RRMS %)	122/45 (73.1)	-	-	-	-	-
EDSS at baseline, median (IQR)	2.5 (1.5–4.5)	1.5 (1.5–2.5)	5.7 (4.0–6.5)	-	-	**<0.001**
EDSS at follow-up, median (IQR)	3.0 (1.5–6.0)	2.0 (1.5–3.4)	6.5 (4.5–6.5)	-	-	**<0.001**
EDSS change, mean (SD)	0.46 (0.9)	0.44 (1.0)	0.49 (0.8)	-	-	0.814
ARR, mean (SD)	0.18 (0.43)	0.20 (0.41)	0.09 (0.3)	-	-	0.115
History of ON, n (%)	84 (50.3)	62 (50.8)	22 (48.8)	-	-	0.688
LCLA 100%, mean (SD)	52.6 (7.9)	53.9 (7.8)	49.3 (7.6)	55.9 (4.5)	**0.001**	**0.002**
LCLA 2.5%, mean (SD)	36.9 (12.9)	38.9 (11.7)	31.9 (14.3)	43.2 (8.6)	**<0.001**	**0.004**
LCLA 1.25, mean (SD)	29.8 (18.1)	31.1 (17.9)	26.5 (18.2)	36.0 (13.9)	**0.018**	0.193
DMT at baseline, n (%)	-					
IFN-β	66 (39.4)	47 (38.5)	19 (42.2)	-	-	**0.038**
GA	33 (19.8)	22 (18.0)	11 (24.4)			
Natalizumab	23 (13.8)	18 (14.8)	5 (11.1)			
Oral DMT	0 (0.0)	0 (0.0)	0 (0.0)			
Off-label DMT	4 (2.4)	2 (1.7)	2 (4.4)			
Not on any DMT	41 (24.6)	33 (27.0)	8 (17.8)			

Legend: pwMS—people with multiple sclerosis, HCs—healthy controls, EDSS—Expanded Disability Status Scale, ARR—annualized relapse rate, ON—optic neuritis, DMT—disease-modifying therapy, IFN—interferon, GA—glatiramer acetate, SD—standard deviation, IQR—interquartile range, LCLA—low-contrast letter acuity measured at 100% contrast, 2.5% contrast, and 1.25% contrast. All LCLA measures are collected at the follow-up visit. Teriflunomide, dimethyl fumarate, and fingolimod were considered oral DMTs. Intravenous immunoglobulins, azathioprine, mitoxantrone, and mycophenolic acid were considered as off-label DMTs. *p*-value lower than 0.05 was considered statistically significant and shown in bold.

**Table 2 diagnostics-13-00596-t002:** Intra- and inter-rater reproducibility of OCT-based vasculature measures.

OCT-Based Measure	Rater 1—Trial 1	Rater 1—Trial 2	Rater 2	Intra-Rater Reproducibility		Inter-Rater Reproducibility	
				ICC	*p*-Value	ICC	*p*-Value
Total vessel diameter	1.19 (0.2)	1.09 (0.2)	0.92 (0.2)	0.906 (0.8–0.956)	**<0.001**	0.774 (0.519–0.894)	**<0.001**
Number of vessels	15.1 (2.5)	16.4 (3.3)	15.1 (2.9)	0.762 (0.493–0.888)	**<0.001**	0.876 (0.736–0.942)	**<0.001**

Legend: OCT—optical coherence tomography, ICC—intraclass correlation coefficient. The raw data are shown as mean (standard deviation), whereas the ICC is shown as type C correlation coefficient (95% confidence intervals for lower and upper bounds). Two-way mixed effects models were used. *p*-value lower than 0.05 was considered statistically significant and shown in bold.

**Table 3 diagnostics-13-00596-t003:** Retinal layer thickness and retinal vasculature measures in the study population at their study visits.

Retinal Layer Thickness and Retinal Vasculature Measures	pwMS		pwRRMS		pwPMS		HCs		pwMS vs. HCs Age Adjusted *p*-Value	pwRRMS vs. pwPMS Age-Adjusted *p*-Value
	*n* = 101	*n* = 167	*n* = 73	*n* = 113	*n* = 28	*n* = 54	*n* = 35	*n* = 48		
Baseline peripapillary RNFLT (µm)	86.3 (12.9)		88.6 (12.1)		79.4 (13.2)		99.9 (9.5)		**0.004**	**0.015**
Follow-up peripapillary RNFLT (µm)	82.8 (12.9)		86.1 (11.6)		74.1 (12.3)		95.9 (11.4)		**<0.001**	**<0.001**
Baseline macular volume (mm^3^)	7.9 (0.9)		7.9 (0.9)		7.6 (0.8)		8.8 (0.6)		**0.004**	0.317
Follow-up macular volume (mm^3^)	8.2 (0.4)		8.3 (0.4)		7.9 (0.4)		8.5 (0.5)		**0.001**	**0.003**
Baseline macular GCIPLT (µm)	75.3 (12.2)		78.2 (9.9)		65.4 (13.7)		86.8 (7.2)		**0.016**	0.095
Follow-up macular GCIPLT (µm)	74.1 (8.3)		75.5 (8.2)		70.4 (7.6)		79.9 (9.3)		**<0.001**	0.176
Baseline total vessel diameter (cm)	2.6 (0.3)		2.60 (0.31)		2.50 (0.4)		2.5 (0.4)		0.082	0.794
Follow-up total vessel diameter (cm)	2.5 (0.4)		2.56 (0.42)		2.43 (0.3)		2.7 (0.3)		0.182	0.899
Baseline number of vessels (n)	36.7 (7.4)		37.14 (7.3)		35.18 (7.6)		34.8 (7.2)		0.987	0.691
Follow-up number of vessels (n)	35.1 (5.9)		35.7 (5.9)		33.6 (5.6)		36.8 (5.3)		**0.017**	0.973
Baseline average vessel diameter (cm)	0.07 (0.01)		0.07 (0.01)		0.07 (0.01)		0.07 (0.01)		0.167	0.975
Follow-up average vessel diameter (cm)	0.074 (0.01)		0.07 (0.1)		0.08 (0.1)		0.075 (0.01)		0.457	0.527

Legend: pwMS—people with multiple sclerosis. pwRRMS—people with relapsing–remitting multiple sclerosis. pwPMS—people with progressive multiple sclerosis. HCs—healthy controls. RNFLT—retinal nerve fiber layer thickness. GCIPLT—ganglion cell inner plexiform layer thickness, n—number. cm—centimeters. µm—micrometers. The N in the first column of each group represents the sample size with available data from the baseline timepoint and the N in the second column represents the sample size with available data from the follow-up timepoint. All values are shown as mean and standard deviation. The sample size is shown for both the baseline and follow-up visits for each group, respectively. *p*-value lower than 0.05 were considered statistically significant and shown in bold.

**Table 4 diagnostics-13-00596-t004:** Longitudinal changes in retinal vasculature over the follow-up.

	pwMS (*n* = 167)	HCs (*n* = 48)	pwRRMS (*n* = 122)	pwPMS (*n* = 45)	pwMS vs. HCs *p*-Value	pwRRMS vs. pwPMS *p*-Value
Change in total vessel diameter (cm), mean (SD)	0.06 (0.5)	0.3 (0.5)	0.08 (0.5)	0.005 (0.6)	**0.017**	0.504
Change in number of vessels (n), mean (SD)	−3.7 (8.4)	0.7 (7.6)	−3.5 (8.9)	−4.4 (7.2)	**0.007**	0.617
Change in average per-vessel diameter (cm), mean (SD)	0.008 (0.02)	0.007 (0.02)	0.007 (0.02)	0.01 (0.02)	0.661	0.482

Legend: pwMS—people with multiple sclerosis, HCs—healthy controls, pwRRMS—people with relapsing–remitting multiple sclerosis, pwPMS—people with progressive multiple sclerosis, cm—centimeter, n—number, SD—standard deviation. The comparison was performed using Student’s t-test. *p*-values lower than 0.05 were considered statistically significant and shown in bold.

**Table 5 diagnostics-13-00596-t005:** Relationship between retinal vasculature and retinal layer thickness in pwMS and HCs at the follow-up visit.

RNFL Associations	pwMS (*n* = 167)		pwRRMS (*n* = 113)		pwPMS (*n* = 54)		HCs (*n* = 48)	
	r-Value	*p*-Value	r-Value	*p*-Value	r-Value	*p*-Value	r-Value	*p*-Value
Total vessel diameter	**0.216**	**0.007**	0.187	0.058	0.238	0.096	0.249	0.099
Number of vessels	**0.191**	**0.018**	**0.221**	**0.025**	0.091	0.53	0.289	0.054
Average vessel diameter	−0.002	0.983	−0.017	0.862	0.095	0.513	−0.218	0.151
**MV Associations**	**pwMS** **(*n* = 167)**		**pwRRMS** **(*n* = 113)**		**pwPMS** **(*n* = 54)**		**HCs** **(*n* = 48)**	
Total vessel diameter	**0.259**	**0.001**	**0.238**	**0.017**	**0.335**	**0.017**	−0.027	0.864
Number of vessels	**0.174**	**0.033**	0.136	0.177	0.22	0.124	0.068	0.668
Average vessel diameter	0.05	0.543	0.066	0.514	0.109	0.45	−0.256	0.102
**GCIPLT Associations**	**pwMS** **(*n* = 167)**		**pwRRMS** **(*n* = 113)**		**pwPMS** **(*n* = 54)**		**HCs** **(*n* = 48)**	
Total vessel diameter	0.112	0.172	−0.013	0.893	**0.387**	**0.007**	0.111	0.462
Number of vessels	**0.252**	**0.002**	0.165	0.096	**0.355**	**0.013**	0.025	0.867
Average vessel diameter	−0.152	0.063	−0.155	0.119	−0.07	0.636	−0.071	0.641

Legend: pwMS—people with multiple sclerosis. pwRRMS—people with relapsing–remitting multiple sclerosis. pwPMS—people with progressive multiple sclerosis. HCs—healthy controls. RNFLT—retinal nerve fiber layer thickness. MV—macular volume, GCIPLT—ganglion cell inner plexiform layer thickness. *p*-value lower than 0.05 was considered statistically significant and shown in bold.

## Data Availability

The data that support the findings of this study are available from the corresponding author, D.J., upon reasonable request.

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
