# Peer review of "Retinal Blood Vessel Analysis Using Optical Coherence Tomography (OCT) in Multiple Sclerosis"

_diagnostics, 2023, doi:10.3390/diagnostics13040596_

Round 1
Reviewer 1 Report
In this study, the authors aimed to determine differences in the retinal vasculature between MS patients and healthy controls (HC), and to determine the relationship between retinal vascular properties and the thickness of the retinal nerve fiber layer and retinal parameters.
The results showed significantly fewer retinal blood vessels in MS compared to HC. During 5.4-year follow-up, the results showed significantly reduced numbers and diameters of the retinal vessels in MS compared to HC.
The major concern about the study is that the results were not presented according to optic neuritis (ON) and non-ON MS subgroup although the authors noted There were no differences in retinal vessel measurements at baseline between MSON and non-MSON eyes. It is thus difficult to interpret whether the changes in retinal vessels in MS are the result of neurodegeneration or as a consequence of the neuroinflammation. Note: in the abstract, the authors describe "increase in total vessel diameter" in MS.
Furthermore, at baseline, both the numbers and diameters of retinal vessels are greater in MS than in controls. The results from 5.4 years of follow-up showed the opposite (Table 3). Moreover, with time, both the number and the diameters of retinal vessels increased in HC, not in MS. How could authors explain these phenomena?
Finally, it is important to present data with scatter plot to show correlation between retinal vasculature data and RNFL, GCIPL, respectively.
The major concern about the study is that the results were not presented according to optic neuritis (ON) and non-ON MS subgroup although the authors noted "There were no differences in retinal vessel measurements at baseline between MSON and non-MSON eyes". It is thus difficult to interpret whether the changes in retinal vessels in MS are the result of neurodegeneration or as a consequence of the neuroinflammation. Note: in the abstract, the authors describe "increase in total vessel diameter" in MS.
Furthermore, at baseline, both the number and diameters of retinal vessels are greater in MS than in controls. The results from 5.4 years of follow-up showed the opposite (Table 3). Moreover, with time, both the number and the diameters of retinal vessels increased in HC, not in MS. How could writers explain these phenomena?
Finally, it is important to present data with point scatter to show correlation between retinal vascular data and RNFL, GCIPL, respectively.
Author Response
Review 1:
In this study, the authors aimed to determine differences in the retinal vasculature between MS patients and healthy controls (HC), and to determine the relationship between retinal vascular properties and the thickness of the retinal nerve fiber layer and retinal parameters.
The results showed significantly fewer retinal blood vessels in MS compared to HC. During 5.4-year follow-up, the results showed significantly reduced numbers and diameters of the retinal vessels in MS compared to HC.
Response: We thank the Reviewer for the suggestions that have improved our manuscript. Point-by-point answers are provided hereafter.
The major concern about the study is that the results were not presented according to optic neuritis (ON) and non-ON MS subgroup although the authors noted There were no differences in retinal vessel measurements at baseline between MSON and non-MSON eyes. It is thus difficult to interpret whether the changes in retinal vessels in MS are the result of neurodegeneration or as a consequence of the neuroinflammation. Note: in the abstract, the authors describe "increase in total vessel diameter" in MS.
Response: As suggested by the Reviewer, we have included the information regarding the MSON and non-MSON eyes. We thank for noticing the mistake in our abstract. This is now corrected. Additional discussion regarding the inflammation effect on vessels in pwMS has been added in the Discussion section.
Furthermore, at baseline, both the numbers and diameters of retinal vessels are greater in MS than in controls. The results from 5.4 years of follow-up showed the opposite (Table 3). Moreover, with time, both the number and the diameters of retinal vessels increased in HC, not in MS. How could authors explain these phenomena?
Response: We thank the Reviewer for this comment. We have further expanded on the potential hypothesis why the changes are as shown from a neuroimmunological point of view. Moreover, a very similar dynamic change (initially higher number that has steep decline) in the global vasculature has been seen between the pwMS vs. HCs.
- Absinta M, Nair G, Monaco MCG, et al. The "central vein sign" in inflammatory demyelination: The role of fibrillar collagen type I. Ann Neurol 2019;85:934-942.
- la Sala A, Pontecorvo L, Agresta A, Rosano G, Stabile E. Regulation of collateral blood vessel development by the innate and adaptive immune system. Trends Mol Med 2012;18:494-501.
- Weel Vv, Toes REM, Seghers L, et al. Natural Killer Cells and CD4+ T-Cells Modulate Collateral Artery Development. Arteriosclerosis, Thrombosis, and Vascular Biology 2007;27:2310-2318.
Finally, it is important to present data with scatter plot to show correlation between retinal vasculature data and RNFL, GCIPL, respectively.
Response: We agree with the Reviewer. We have provided scatter plots for the significant correlations between retinal vasculature data, RNFL and GCIPL. This is now part of the manuscript as Figure 3.
Reviewer 2 Report
Please see the attached document.

Author Response
Reviewer 2:
This study focuses on Retinal Blood Vessel Analysis Using Optical Coherence Tomography (OCT) in Multiple Sclerosis (MS) vs healthy controls and determine the relationship with structural neuroretinal thickness (specially in peripapillary retinal nerve fiber layer and macular ganglion cell inner plexiform layer). Patients with MS showed fewer retinal blood vessels and suffered a decrease in number of vessels but an increase in vessel diameter over 5 years of follow-up. RNFL atrophy was associated with the vasculature outcomes.
- General concept comments
The manuscript scientifically sound interesting since it explores the retinal vasculature in patients with MS. The article is generally well written, is relevant for the field and presented in a well-structured manner.
The experimental design is technically doubtful to test the hypothesis. The results are meaningful, but a little bit extensive and unattractive in the form that they are shown. At this version of the manuscript results would not be reproducible based on some missing details in the methods section. The discussion is well organized, clarifying and sufficient. Conclusions are consistent with the evidence and arguments presented. The references are adequate, sufficient and in general, meaningful.
However, the manuscript is not acceptable in its present form, for further consideration some issues should be improved. In my opinion methodology is the main point that need to be improved to considered the results scientifically confident. I found there are a number of major methodological concerns, some errors or missing data that compromised the rigor.
Response: We thank the Reviewer for the excellent review and very constructive commentary. We have addressed all comments and provide point-by-point responses hereafter. The review has significantly improved the quality of the manuscript.
- Specific comments I would suggest the following:
ABSTRACT:
- Line 16. Please define MS for first mention.
Response: This has been expanded.
2) Line 20. Vascular analysis is made in peripapillary RNFL. I would suggest to change retinal vasculature characteristics to neuroretinal or peripapillary .
Response: This has been added as suggested.
3) Line 23. Could you define OCTSEG please?
Response: This has been expanded as suggested.
4) Lines 26-27. Correlations are both positive?? Or the first is positive and the second negative?
Response: Both correlations are positive. This has been further clarified in the abstract.
5) Lines 27-28: This sentence sounds confounding to me. The per-vessel diameters means inner lumen?
Response: This has been corrected. The vessel diameter includes both the lumen and the vessel wall.
KEYWORDS:
- Please, define OCT and eliminate RNFL.
Response: This has been corrected.
INTRODUCTION:
- Line 36. Please eliminate CNS, since no more times this word is written in the text.
Response: This has been removed.
- Line 45 pwMS. Please define first mention.
Response: This has been added.
- Line 59. ON. Please define first mention.
Response: This has been added.
MATERIALS AND METHODS
Please could you explain and include in method how did you recognize vessels? I consider crucial to explain deeper how the authors get the measurements of vessels, as all the ongoing results are based on that. Vessels are not rectangles but circles, how do to infer measures to circles?
Response: This was further explained in the Materials and Methods.
Could you include the parameters analyzed, not only regarding OCT, but also demographic and clinical data or treatments as you then show in results. The authors mention them but not explain them.
Response: These are now included in the Materials and Methods section.
- Line 81. Please define CEG-MS.
Response: We have clarified the sentence regarding CEG-MS.
- Line 89. Please include the structured interview-based questionnaire.
Response: We have included the reference that provides further details regarding the study questionnaire.
Jakimovski, D.; Gandhi, S.; Paunkoski, I.; Bergsland, N.; Hagemeier, J.; Ramasamy, D.P.; Hojnacki, D.; Kolb, C.; Benedict, R.H.B.; Weinstock-Guttman, B.; et al. Hypertension and heart disease are associated with development of brain atrophy in multiple sclerosis: a 5-year longitudinal study. European Journal of Neurology 2019, 26, 87-e88, doi:https://doi.org/10.1111/ene.13769.
- Line 92. The authors assessed the vision abilities only at the follow-up time point. Why?
Response: The study did not incorporate the LCLA test since the start of the study. Later on, this omission was corrected and the test was added for the follow-up period.
Line 95. Could the authors include the code of the IRB, please?
Response: This was already included in the IRB statement as part of the MDPI template. “The study was conducted in accordance with the Declaration of Helsinki, and approved by the Institutional Review Board of University at Buffalo (protocol code 00006278 approved on 4/29/2022).”
- OCT analysis: the right or left eye was considered? I think it is very important to show if you always analyzed the right eye or the left or the proportion of each; and if you found differences between eyes, since differences in vascularization were described. Did you consider the refractive error, since high myopic or hyperopic patients could have retinal magnification.
Response: We thank the Reviewer for this comment. The OCT scan routinely corrects for the refractive error before images are being taken. This is part of the standard operating procedure in OCT acquisition. This has been clarified in the manuscript. We looked into the inter-eye difference between and this was included in the manuscript.
6) Line 98: pwPMS. Please define first mention. The patients with Progressive MS, they were primary or secondary progressive?
Response: This was defined. The classification and reference for the Lublin criteria was added. We also included in the methods that the secondary-progressive and primary-progressive were merged into one group. We also outlined how many were from each sub-category.
- Lublin FD, Reingold SC, Cohen JA, et al. Defining the clinical course of multiple sclerosis: the 2013 revisions. Neurology. 2014;83(3):278-286. doi:10.1212/WNL.0000000000000560
- Line 99: “Baseline OCT scans were retrieved from previous patient timepoints” Which ones??
Response: We thank the Reviewer for noticing this mistake. It was corrected in the manuscript. It referred to the timepoint from the baseline study visit.
- Line 104. Please include if the retinal vasculature segmentation on the peripapillary RNFL scans and its posterior analysis belongs to superficial, deep plexus…?
Response: We have added that all vessels segmented in this manuscript correspond to the superficial capillary plexus.
- Line 105. There is a typing error; cicrle diameter.
Response: This was corrected.
- Line 108. Could you define GUI or add an appropriate reference?
Response: We expanded the common abbreviation for Graphical User Interface. We have included the link directly to the software.
- Line 109. An important handicap of the study is the manual segmentation. Although I acknowledge the effort made by the authors to avoid bias and intra- and interobserver analysis because their methodology requires manual analysis.
Response: We have re-emphasized the semi-automated nature of the analysis in the Discussion of the manuscript.
- Line 118. Please could you define IDE?
Response: We have expanded the IDE abbreviation.
- Line 121. Please explain in much more detail how the measurements are made as they are not very clear. The diameter is circular and the image shown in figure 1 is rectangular. Can you please tell, or even better, show in image how you identify the circle? Could you also calculate the area of that diameter?. I encourage to explain the methodology as other authors previously did.
Bhaduri B, Nolan RM, Shelton RL, Pilutti LA, Motl RW, Moss HE, Pula JH, Boppart SA. Detection of retinal blood vessel changes in multiple sclerosis with optical coherence tomography. Biomed Opt Express. 2016 May 20;7(6):2321-30. doi: 10.1364/BOE.7.002321. PMID: 27375947; PMCID: PMC4918585.
Response: We have included the recommended reference to our methods as well. The methods in segmentation were identical, given that the same software was used in both studies. We have further explained the measurement in our manuscript and added a new Figure that explains the Reviewers question.
14) Line 121. GCIP. I would suggest to use the acronim GCIPL with the L of layer trhoughout the paper, and please define it as this is first mention.
Response: This has been corrected as suggested.
15) Please why did you choose GCIP in macular area and why peripapillary RNFL?
Response: This is due to the nature of how images were taken. Typically, in the field of OCT use in MS, the RNFL is measured on a A-scan peri-papillary image. On the other hand, the macular site is typically used for volume-based (B scan) images which allow quantification of each layer. Since the implementation of full layer analysis from Heidelberg (this occurred over the past several years) we have resorted of acquiring B scans of the peri-papillary site as well. For this particular analysis we do not have B-scans available.
This was added in the limitation of the manuscript as well.
RESULTS
Did the author find a particular sector o sector more affected? Did you consider analyzing by peripapillary sectors? It would be very interesting to include it.
Response: We thank the Reviewer for this interesting proposition. We agree that this additional analysis would be valuable and will incorporate it in our new manuscript that intends at investigating the change in retinal vasculature and global brain perfusion measures in multiple sclerosis patients. From an initial perspective, the sectors are anatomically highly variable among people and any findings could be heavily compounded by this factor.
1) Line 165. Table 1. Please use the same number of decimals along the table. I would suggest using always two decimals.
Response: We have shortened to two decimals in all fields of Table 1.
- Why the row “time of follow-up, mean (SD) is empty?.
Response: We apologize for this omission. This was corrected and added in the table.
- INF beta 49 + 19 are not 66. Is it correct?, Off label DMT 0+ 2 is not 4. Is it correct?, p= 0.038 should be in bold.
Response: We apologize for this error. This was corrected in Table 1.
- Line 189. There is an error. I think it should be pwMS instead of HCs.
Response: We thank the Reviewer for noticing the mistake. This is corrected.
- Line 194. Could you use superscript numbers please?
Response: We have used the superscript for mm3 throughout the manuscript.
- Line 197. I could not find this information in the table, could you include it at least in the text please?
Response: As suggested, we incorporated the MSON vs. non-MSON data in the text of the Results.
- Line 199. Table 4 is confounding for me. This table 4 does not show the results written in text. please check, it is an increase or a decrease in vessel diameter, total or average? Does the change in average vessel diameter add some information? Please if so include it.
Response: We thank the Reviewer for point out this change. We have further explained the measures in Table 4 and added it in the text. The change in average per-vessel diameter was not significant.
- Table 3: the first N belongs to the Retinal layers? and the second N to retinal vasculature measures? Please could you clarify it?. Could you include that the RNFL is peripapillary please?, Please use the same number of decimals. I would suggest using always two decimals.
Response: We have further clarified the sample size in Table 3. The first N belongs to the baseline and the second N belongs to the follow-up sample size. This added in the Legend. We have corrected the decimals and added peripapillary as well.
- Line 223, all these correlations are significant but very low…I would suggest to include this “handicap” in the discussion section.
Response: We agree with the Reviewer. This was added in the Discussion section.
- Line 240. Table 5. In my opinion there are many tables in the manuscript, that made reading unattractive. For making it more attractive, perhaps showing correlations in graphs would improve it.
Response: We agree with the Reviewer. We have provided scatter plots for significant correlations between retinal vasculature data, RNFL and GCIPL. This is now part of the manuscript as Figure 3.
DISCUSSION
- Line 292. I miss a reference here.
Response: We have included a reference on this sentence.
- Qureshi SS, Beh SC, Frohman TC, Frohman EM. An update on neuro-ophthalmology of multiple sclerosis: the visual system as a model to study multiple sclerosis. Curr Opin Neurol 2014;27:300-308.
REFREENCES
- Line 65. Reference 9. I could not find it. Is it correct?
Response: This reference is a comprehensive chapter of MRI imaging in MS. Here is the link of the chapter for that particular reference:
https://link.springer.com/chapter/10.1007/978-3-030-24436-1_6
- Line 81. Reference 12 does not sound correct.
Response: The Reference 12 was correct, but we removed it as the Reviewer has suggested. However, in order to further clarify the CEG-MS study, we have included two different more reference that utilized data from the same cohort of pwMS and HCs.
- Jakimovski D, Zivadinov R, Vaughn CB, Ozel O, Weinstock-Guttman B. Clinical effects associated with five-year retinal nerve fiber layer thinning in multiple sclerosis. J Neurol Sci 2021;427:117552.
- Tavazzi E, Jakimovski D, Kuhle J, et al. Serum neurofilament light chain and optical coherence tomography measures in MS: A longitudinal study. Neurol Neuroimmunol Neuroinflamm 2020;7.
- Line 290. Reference 23. This is an arvo meeting abstract, I would suggest to use another reference more robust.
Response: Unfortunately, we were not able to find a reference for a full article on this work. The Authors of that work may have not published it in an indexed journal.
Round 2
Reviewer 2 Report
Attached document.

Author Response
Reviewer 2: The authors have substantially improved the article by answering the questions raised. However, there are still some issues that need to be improved in order to publish the article, as there are currently some discrepancies that make publication impossible.
Response: We thank the Reviewer for the detailed review that significantly improves the quality of our work. We are highly appreciative of the attention to details and we have addressed all comments hereafter.
1) Lines 27-28 old version: This sentence sounds confounding to me. The per-vessel diameters means inner lumen? Response: This has been corrected. The vessel diameter includes both the lumen and the vessel wall. Line 26 new version: “Moreover, the total vessel diameter in pwMS did not change when compared to and a significant the increase in vessel diameter in the HCs (0.06 vs. 0.03, p=0.017). …” Please check these numbers as they say the opposite and in the results section of the manuscript the authors have written 0.06 vs 0.3. And on line 254 it is written the opposite “PwMS showed a significantly greater increase in vessel diameter and a significant reduction in the number of retinal blood vessels when compared to HCs (0.06 cm vs. 0.3 cm, p = 0.017 and -3.7 vs. 0.7, p = 0.007, respectively).” Please correct it.
Response: We have corrected the numbers as recognized. It is 0.06 vs. 0.3, p=0.017.
Moreover, this sentence keeps on sounding confounding to me: “The 5-year RNFL atrophy was associated with the decrease of number of retinal vessels (p=0.045), per-vessel total diameter which includes both the lumen and vessel wall (p=0.025) and increase in total vessel diameter (p=0.009).” How is it possible to decrease of number of retinal vessels and per-vessel total diameter but to increase in total vessel diameter?.
Response: We agree with the Reviewer. We further investigated this discrepancy and looked at the data case by case. The directionally was correctly reported. That said, the change in number of vessels and total vessel diameter had very low correlation (r=0.295). Many pwMS had 10+ decrease in number of vessels and stable total diameter (increase in compensatory per vessel diameter), while on the other hand, pwMS could have an increase in the number of vessels and decrease in total diameter (more but smaller vessels). When these pwMS are averaged as part of a group comparison, the differences would be nullified. Development of change patters that could incorporate different directionality changes could improve the overall associations between retinal vasculature measures with clinical and phenotypical outcomes. This is now included in the manuscript.
Therefore, we have removed the sentence from the abstract and left the findings in the Results section of the manuscript. Further limitations regarding the variability between the pwMS and their retinal vasculature is included in the manuscript.
2) Line 83 new version: Please could you change “prospective study that examined cardiovascular, genetic, and environmental factors in multiple sclerosis (CEG-MS)” to “prospective study that examined cardiovascular, environmental and genetic factors in multiple sclerosis (CEG-MS)”?
Response: This has been corrected.
3) Line 98 old version: pwPMS. The patients with Progressive MS, they were primary or secondary progressive? Response: This was defined. The classification and reference for the Lublin criteria was added. We also included in the methods that the secondary-progressive and primary-progressive were merged into one group. We also outlined how many were from each sub-category. Lublin FD, Reingold SC, Cohen JA, et al. Defining the clinical course of multiple sclerosis: the 2013 revisions. Neurology. 2014;83(3):278-286. doi:10.1212/WNL.0000000000000560 Although authors affirm that in this new version of the manuscript they included in methods “that the secondary-progressive and primary-progressive were merged into one group. And that outlined how many were from each sub-category”. It was impossible to me to find this information, please, could you clarify it?
Response: We thank the Reviewer for the comment. We further included the statement and number of SPMS and PPMS patients that were part of the PMS group.
4) Line 165 old version. Table 1. Please use the same number of decimals along the table. I would suggest using always two decimals. Response: We have shortened to two decimals in all fields of Table 1. Table 1 sometimes shows 1 decimal and sometimes 2 decimals, please use always the same number of decimals.
Response: We have corrected the Table 1 as suggested.
5) Line 185 new version: please write p<0.001 instead of p=<.001, in “(median 5.7 vs. 1.5, p = <.001 and median 6.5 vs. 2.0, p = <.001, respectively).
Response: This has been corrected.
6) Line186 new version: table 1 shows the opposite to text “There was also significant difference in the history of optic neuritis or disease modifying therapies amongst pwRRMS and pwPMS. “ Please correct it, is it difference in ON history or not?
Response: We have further corrected the sentences. The MSON was not different between the groups (p=0.688).
7) Line 191 new version: there is a typing error: worse vision instead of worst vision?
Response: This has been corrected.
8) Line 228 new version: Please, correct p=354.
Response: This has been corrected.
9) Line 197 old version. I could not find this information in the table, could you include it at least in the text please? Response: As suggested, we incorporated the MSON vs. non-MSON data in the text of the Results. Line 233-235 new version: there is lack of p. Include it. “When compared to the 214 non-MSON eyes, the 120 MSON-affected eyes of pwMS had similar per-eye total vessel diameter (1.2 vs. 1.3, p=), similar average number of vessels (17.2 vs. 17.7, p=) and similar average per-vessel average diameter (0.07 vs. 0.07, p=0.)”.
Response: We apologize for the omission. This has been included in the new revised version of the manuscript.
10) Line 199 old version. Table 4 is confounding for me. This table 4 does not show the results written in text. please check, it is an increase or a decrease in vessel diameter, total or average? Does the change in average vessel diameter add some information? Please if so include it. Response: We thank the Reviewer for point out this change. We have further explained the measures in Table 4 and added it in the text. The change in average per-vessel diameter was not significant. I feel it is not correct yet. Authors wrote “PwMS showed a significantly greater increase in vessel diameter and a significant reduction in the number of retinal blood vessels when compared to HCs (0.06 cm vs. 0.3 cm, p = 0.017 and -3.7 vs. 0.7, p = 0.007, respectively).” However 0.06 is lower than 0.3…
Response: We apologize for the omission. This has been corrected accordingly. The pwMS have smaller increase in vessel diameter than the HCs.
11) Line 253 new version: Please add vasculature.
Response: This has been corrected.
12) Table 5 please, include MV definition in the legend.
Response: This has been corrected.
13) Line 278 new version: Please add an extra L in GCIP.
Response: This has been corrected.